# tRNA synthetase counteracts c-Myc to develop functional vasculature

Yi Shi[1,2], Xiaoling Xu[1,2†], Qian Zhang[1,2], Guangsen Fu[1,2], Zhongying Mo[1,2], George S Wang[1,2], Shuji Kishi[3], Xiang-Lei Yang[1,2]*

[1]Department of Chemical Physiology, The Scripps Research Institute, La Jolla, United States; [2]Department of Cell and Molecular Biology, The Scripps Research Institute, La Jolla, United States; [3]Department of Metabolism and Aging, The Scripps Research Institute, Jupiter, United States

**Abstract** Recent studies suggested an essential role for seryl-tRNA synthetase (SerRS) in vascular development. This role is specific to SerRS among all tRNA synthetases and is independent of its well-known aminoacylation function in protein synthesis. A unique nucleus-directing domain, added at the invertebrate-to-vertebrate transition, confers this novel non-translational activity of SerRS. Previous studies showed that SerRS, in some unknown way, controls VEGFA expression to prevent vascular over-expansion. Using in vitro, cell and animal experiments, we show here that SerRS intervenes by antagonizing c-Myc, the major transcription factor promoting VEGFA expression, through a tandem mechanism. First, by direct head-to-head competition, nuclear-localized SerRS blocks c-Myc from binding to the *VEGFA* promoter. Second, DNA-bound SerRS recruits the SIRT2 histone deacetylase to erase prior c-Myc-promoted histone acetylation. Thus, vertebrate SerRS and c-Myc is a pair of 'Yin-Yang' transcriptional regulator for proper development of a functional vasculature. Our results also discover an anti-angiogenic activity for SIRT2.

*For correspondence: xlyang@scripps.edu

Present address: †Institute of Ageing Research, School of Medicine, Hangzhou Normal University, Hangzhou, China

Competing interests: The authors declare that no competing interests exist.

## Introduction

In vertebrates from fish to humans, the vasculature is one of the most important and earliest networks to develop. Surprisingly, three independent forward genetics studies in zebrafish suggested an essential role for seryl-tRNA synthetase (SerRS) in vascular development (*Amsterdam et al., 2004*; *Fukui et al., 2009*; *Herzog et al., 2009*). In fish embryos, disruption of *sars* (gene encoding SerRS), through insertional mutagenesis (*Amsterdam et al., 2004*) or ENU mutagenesis-induced point/truncation mutations (*Fukui et al., 2009*; *Herzog et al., 2009*), caused excessive and abnormal blood vessel growth.

As a member of the aminoacyl-tRNA synthetases family, SerRS is well-known for its essential function in aminoacylation of tRNA[Ser] for protein synthesis in the cytoplasm. However, the role of SerRS in vascular development is independent of its enzymatic activity (*Fukui et al., 2009*), but dependent on its vertebrate-specific, non-catalytic, C-terminal domain UNE-S (*Guo et al., 2010*; *Xu et al., 2012*; *Guo and Schimmel, 2013*). The UNE-S domain contains a robust nuclear localization signal (NLS) sequence that, at least in human cells, directs a substantial amount of cellular SerRS into the nucleus (*Guo et al., 2010*; *Xu et al., 2012*; *Guo and Schimmel, 2013*). Remarkably, all non-null mutations of *sars* linked to vasculature abnormalities in the aforementioned genetics studies either have the NLS truncated or conformationally sequestered, and thus render deficient SerRS nuclear localization (*Xu et al., 2012*). Conversely, zebrafish expressing engineered catalytically active but NLS-mutated SerRS exhibited the same abnormal blood vessel phenotype as observed in the *sars* mutant embryos (*Xu et al., 2012*). Therefore, it has been clearly established that the essential role of SerRS in vascular development arises from its evolutionarily acquired nuclear presence.

**eLife digest** The network of blood vessels is one of the earliest structures to develop in a vertebrate embryo. A protein called Vascular Endothelial Growth Factor A (or VEGFA for short) is needed to promote the growth of these blood vessels, but too much VEGFA can cause blood vessels to grow too much and to grow abnormally.

Like most of the DNA in the nucleus, the gene for VEGFA is tightly wrapped around proteins called histones and must be unwrapped before it can be expressed as a protein. For the *VEGFA* gene, this unwrapping process starts when a protein called c-Myc adds chemical tags to the histones.

Recent research suggested that an enzyme called seryl-tRNA synthetase (or SerRS for short) also controls the expression of VEGFA. This came as a surprise because no other tRNA synthetase has a similar role during development. And although SerRS is known to enter the cell nucleus in vertebrates, researchers did not know what SerRS did in the nucleus to control the expression of VEGFA.

Now, Shi et al. have discovered that SerRS controls blood vessel development in zebrafish embryos by counteracting the activity of c-Myc. It does this in two different ways: first, it directly blocks c-Myc from binding to and unpacking the DNA; and second, SerRS works with another enzyme to remove tags that are already on the histones. Shi et al. found that if the expression of this other enzyme (called SIRT2) was reduced in zebrafish, the fish expressed more VEGFA and their blood vessels grew too much.

Since blood vessel growth is important in the development of cancers, the findings of Shi et al. could also lead to a better understanding of how tumors develop, as well as how blood vessels develop normally.

Interestingly, the vascular abnormalities associated with deficient SerRS nuclear localization were found to be accompanied with a high level of *vegfa* (Vascular Endothelial Growth Factor A) transcript in the mutant fish embryos (*Fukui et al., 2009*; *Xu et al., 2012*). This observation suggested that the nuclear function of SerRS in zebrafish is linked to attenuating the expression of Vegfa. However, the mechanism of the SerRS function has remained obscure. Because VEGFA is a key stimulator of vasculogenesis and angiogenesis for all vertebrates, and over-expression of VEGFA is not only associated with developmental vascular abnormalities, but also contributes to various diseases including cancer (*Drake and Little, 1995*), we were motivated to determine whether the VEGFA-regulating function of SerRS is conserved in higher vertebrates such as humans, and what is the mechanism by which nuclear SerRS controls VEGFA expression.

It is well established that c-Myc is the major transcription factor promoting *VEGFA* gene expression in the nucleus, and thereby has a key role in vascular development. As a basic helix-loop-helix-leucine zipper (bHLHZ) protein, c-Myc functions through heterodimerization with the small bHLHZ partner MAX for binding to the Enhancer Box (E-box) DNA sequence (5'-CACGTG-3') on its target genes (*Blackwood and Eisenman, 1991*). DNA-bound c-Myc recruits histone acetyltransferase to acetylate histone proteins to allow chromatin expansion and activate transcription (*Grandori et al., 2000*). c-Myc knockout mice are embryonic lethal and exhibit, among others deformities, under-developed vasculature. Importantly, these deformities can be partially rescued by transgenic VEGFA expression (*Baudino et al., 2002*). On the other hand, endothelial-specific c-Myc overexpression in mice also causes embryonic lethality arising from widespread edema, multiple hemorrhagic lesions and severe defects in the vascular network, accompanied by an elevated level of VEGFA (*Kokai et al., 2009*). Together, these results suggest that the role of c-Myc in vascular development and in promoting VEGFA expression has to be tightly balanced.

In the work described below, we have elucidated a novel mechanism by which this balance is achieved. We show a head-to-head competition between SerRS and c-Myc for the same *VEGFA* promoter binding site and, in addition, a direct recruitment of SIRT2 histone deacetylase by SerRS to erase the transcription-enhancing chromatin remodeling already instigated by c-Myc. These results and further experiments in a vertebrate model organism reveal that SerRS is a key balancing antagonist of c-Myc for regulation of VEGFA expression, as well as for proper development of a functional vasculature. In addition, our study provides the first report of an anti-angiogenic function for SIRT2, which arises at least in part through its interaction with SerRS.

## Results

### SerRS affects VEGFA expression and angiogenesis in human cells

To study the mechanism of how nuclear SerRS represses VEGFA expression, we started by using human cells. A short hairpin RNA (shRNA) targeting the 3′ UTR of human SerRS mRNA was generated to knock down endogenous SerRS expression in human umbilical vein endothelial cells (HUVECs) and HEK 293 cells (*Figure 1—figure supplement 1A,B*). For both cell types, the level of *VEGFA* transcript was more than doubled in cells expressing the shRNA against SerRS (sh-SerRS) vs a control shRNA (sh-Con) (*Figure 1A*, *Figure 1—figure supplement 2*). Considering that SerRS is an essential component of the translation machinery and that knockdown of SerRS would have a general effect on protein synthesis that may obscure the effect on VEGFA expression, we compensated the 'knockout' cells by expression of NLS-deleted SerRS (ΔNLS) that is fully active in aminoacylation but lost the ability to enter the nucleus (*Xu et al., 2012*), or by expression of wild-type (WT) SerRS (as a separate control). Remarkably, compared to WT SerRS-expressing cells, cells expressing ΔNLS SerRS resulted in three and fourfold higher expression of VEGFA in HUVECs (*Figure 1A*) and HEK293 cells (*Figure 1—figure supplement 2*), respectively. This result supports the idea that the role of nuclear SerRS in suppressing VEGFA expression is conserved from fish to humans. In addition, as measured in an in vitro endothelial tube formation assay, and consistent with the role of VEGFA in promoting angiogenesis, HUVECs expressing ΔNLS SerRS showed a much stronger propensity (than WT SerRS-expressing cells) to form a blood vessel-like tubular network (*Figure 1B–D*).

### SerRS directly binds to the *VEGFA* promoter

Using cellulose beads-linked calf thymus DNA, we found that purified SerRS, but not two other human tRNA synthetases (GlyRS and LysRS), bound to DNA (*Figure 2—figure supplement 1A*). Given this inherent capacity of SerRS to bind DNA, we performed a chromatin immunoprecipitation (ChIP) experiment with 10 primer pairs designed to scan the human *VEGFA* gene, from 4 kb upstream (−4 kb) to 4 kb downstream (+4 kb) of the transcription start site (*Figure 2A,B*). We found that ectopically expressed SerRS bound to the promoter in the region from −1.5 kb to +1 of the start site. Importantly, this region encompasses the binding site of c-Myc on the *VEGFA* promoter (*Figure 2B*; *Kim et al., 2007*).

To investigate whether the binding is repressive in nature, we performed a luciferase assay in which the majority of this promoter region (−1262 ~ +46) was put in front of a luciferase reporter gene to test the transcriptional activity of SerRS. Strikingly, SerRS overexpression sharply reduced the elicitation of luciferase activity (*Figure 2C*, *Figure 2—figure supplement 1B*). In contrast, overexpression of GlyRS, which lacks DNA-binding capacity, had no effect (*Figure 2C*).

In further work, we showed that the inhibitory effect of SerRS persisted as the promoter region was shortened to −262 ~ +46, suggesting that the SerRS-specific responsive element is within this 308-bp region (*Figure 2C*). This DNA fragment was then radiolabeled and subjected to DNase I footprint analysis to further determine the exact SerRS binding sites. As shown in *Figure 2D*, purified SerRS protected a 25-bp region (−62 ~ −38) from DNase I digestion and did so in a concentration dependent manner.

The direct interaction of SerRS with a slightly extended 27-bp DNA fragment (−62 ~ −36) was confirmed by an electrophoretic mobility shift assay (EMSA) (*Figure 3A*). The binding affinity ($K_d$) of SerRS was 211.5 nM as measured by EMSA (*Figure 3A,B*) and 265 nM by surface plasmon resonance (*Figure 3—figure supplement 1*). Interestingly, truncations from both ends of the 27-bp DNA fragment weakened the interaction and, based on the EMSA analysis, the DNA minimal binding site of SerRS was determined to be 21 nt (−59 ~ −38) (*Figure 3C,D*).

### Characterize the interaction between SerRS and DNA

As mentioned above, c-Myc plays a pivotal role in vascular development by promoting VEGFA expression (*Baudino et al., 2002*; *Kokai et al., 2009*). In complex with its partner MAX, c-Myc directly binds to classic or nonclassic E-box sequences on DNA (*Blackwell et al., 1993*; *Kim et al., 2008*) and recruits histone acetyltransferase to allow chromatin expansion and activate transcription. However, the exact binding site of c-Myc/MAX on the *VEGFA* promoter has not been reported. Using DNase I footprint analysis, we also identified the exact c-Myc/MAX binding region on the *VEGFA* promoter. The c-Myc/MAX binding site (−53 ~ −38) contains a nonclassical E-box sequence ([−49]CATGCG[−44]) that completely overlaps with the SerRS binding site (*Figure 2D*).

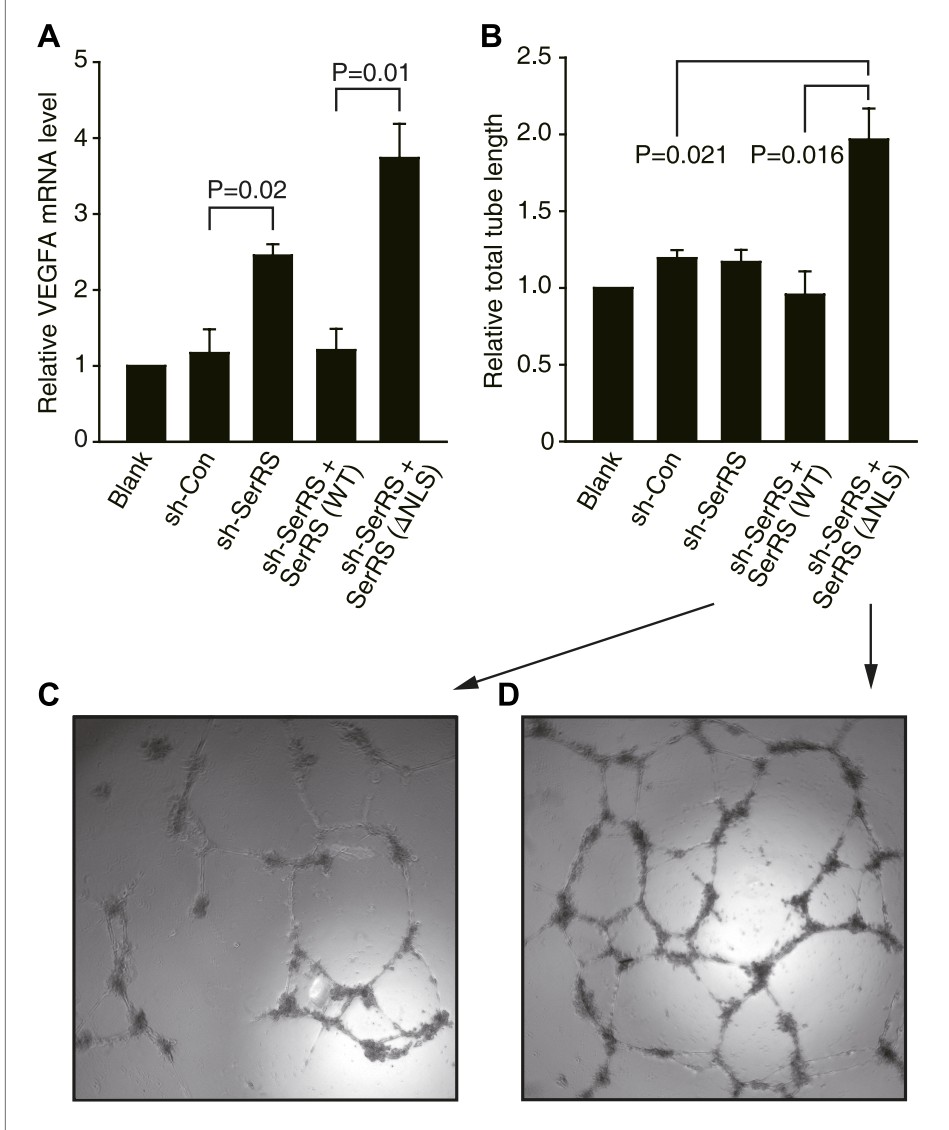

**Figure 1**. Nuclear SerRS suppresses VEGFA expression and angiogenesis. (**A**) VEGFA mRNA levels as detected by real-time RT-qPCR in HUVECs infected with lentiviral plasmids expressing nonspecific control shRNA (sh-Con), SerRS-specific shRNA (sh-SerRS), or sh-SerRS and wild type (WT) or NLS-deleted (ΔNLS) SerRS simultaneously. Values are means ± SEM (n = 3). (**B**) Endothelial tube formation assay to show that excluding SerRS from the nucleus promotes angiogenesis. Values are means ± SEM (n = 3). (**C** and **D**) Representative images of the tubular network formed by HUVECs expressing WT and ΔNLS SerRS, respectively.

The following figure supplements are available for figure 1:

**Figure supplement 1**. Manipulations of the expression of SerRS in HUVEC and HEK 293 cells.

**Figure supplement 2**. Nuclear SerRS suppresses VEGFA expression in HEK 293 cells.

To investigate the sequence-specificity of SerRS in DNA binding and the importance of the E-box sequence for SerRS binding, we designed 11 single or double mutations in the 27 bp DNA, including 5 in the E-box sequence (**Figure 4A**). Two double mutants of the E-box, including one ($^{-49}$CTTACG$^{-44}$) that would completely abolish c-Myc/Max binding (**Blackwell et al., 1993**), did not affect the SerRS interaction (**Figure 4A**); on the other hand, five different single mutations outside the E-box (on both the 5′ and the 3′ sides) that would not affect c-Myc/Max binding greatly weakened SerRS binding (**Figure 4A**), indicating that SerRS and c-Myc/Max have distinct DNA binding specificities.

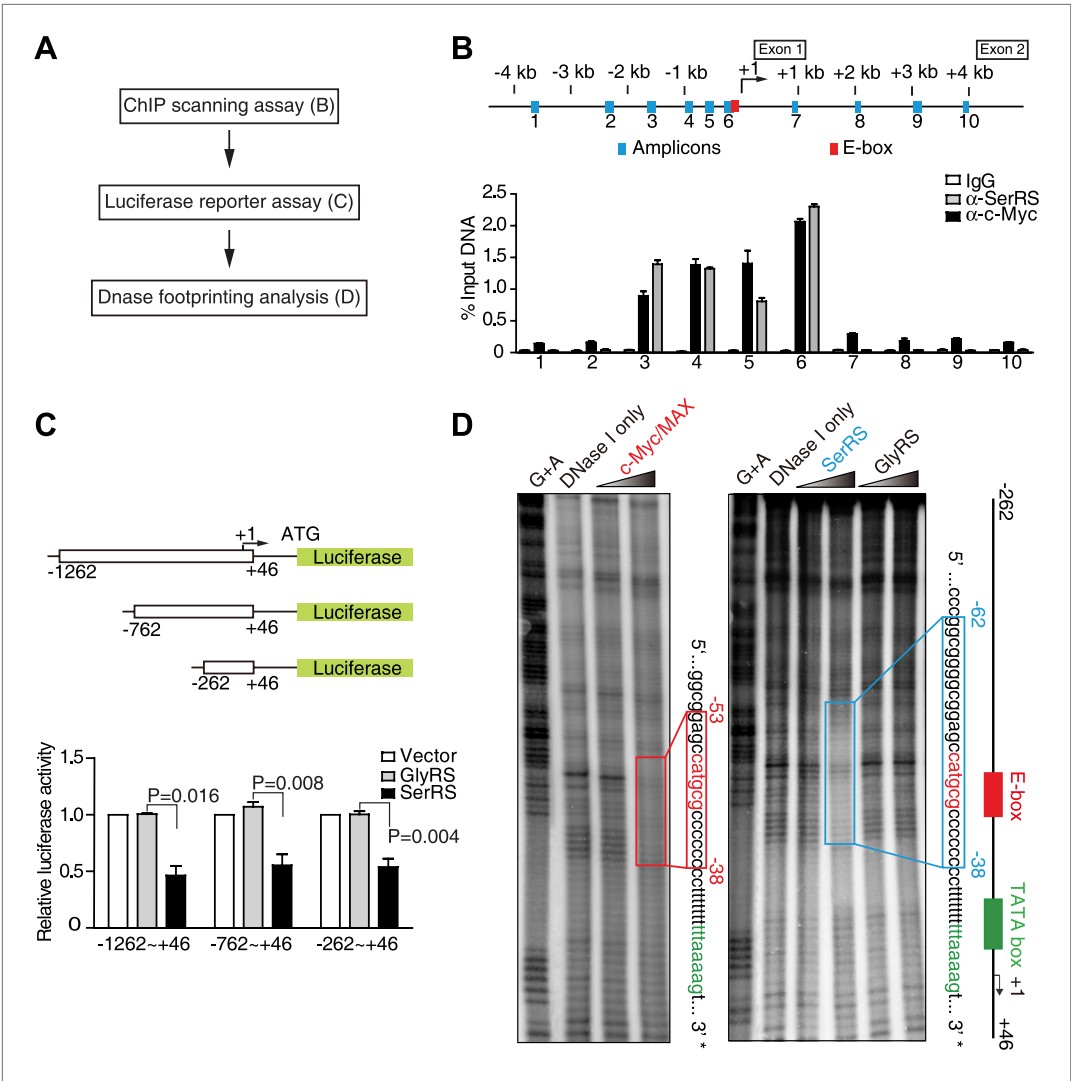

**Figure 2**. Identification of SerRS and c-Myc binding sites on the *VEGFA* promoter. (**A**) Flow chart of consecutive methods used for determining the SerRS binding site. (**B**) Chromatin immunoprecipitation (ChIP) scanning assay to probe the SerRS and c-Myc binding sites. The promoter region of the *VEGFA* gene scanned by 10 amplicons is shown on the top. The amounts of DNA immunoprecipitated by anti-SerRS or anti-c-Myc antibodies or by control IgG from HEK 293 cell lysates were measured by real-time quantitative PCR at each amplicon. The results are represented as percentages of the total input of the chromatin DNA and shown as means ± SEM (n = 3). (**C**) Luciferase assay to confirm the repressive activity of SerRS and narrow down the SerRS binding site on the *VEGFA* promoter. Three different lengths of the *VEGFA* promoter were used to drive luciferase expressions in HEK 293 cells transfected with plasmids expressing SerRS, GlyRS or empty vector. The normalized luciferase activities are shown as mean ± SEM (n = 3). (**D**) In vitro DNase I footprint assay to identify the SerRS binding site. A 308-bp DNA fragment (−262 ~ +46 on the *VEGFA* promoter) radiolabeled at the 3' end was incubated with purified recombinant c-Myc/MAX (1:1 molar ratio), SerRS or GlyRS each at 1 or 5 µM, and then subjected to DNase I digestion. The regions protected by c-Myc/MAX and by SerRS are indicated in red and blue boxes, respectively.

The following figure supplements are available for figure 2:

**Figure supplement 1**. Identification of the interaction between SerRS and DNA.

We also investigated the DNA binding sites on SerRS through domain mapping and deletion mutagenesis. SerRS functions as a dimer in aminoacylation and the dimerization interface is mediated through the catalytic domain (CD). The N-terminal tRNA binding domain (TBD) of SerRS is used for recognizing the long variable arm of tRNA^Ser, while the C-terminal UNE-S domain directs SerRS into

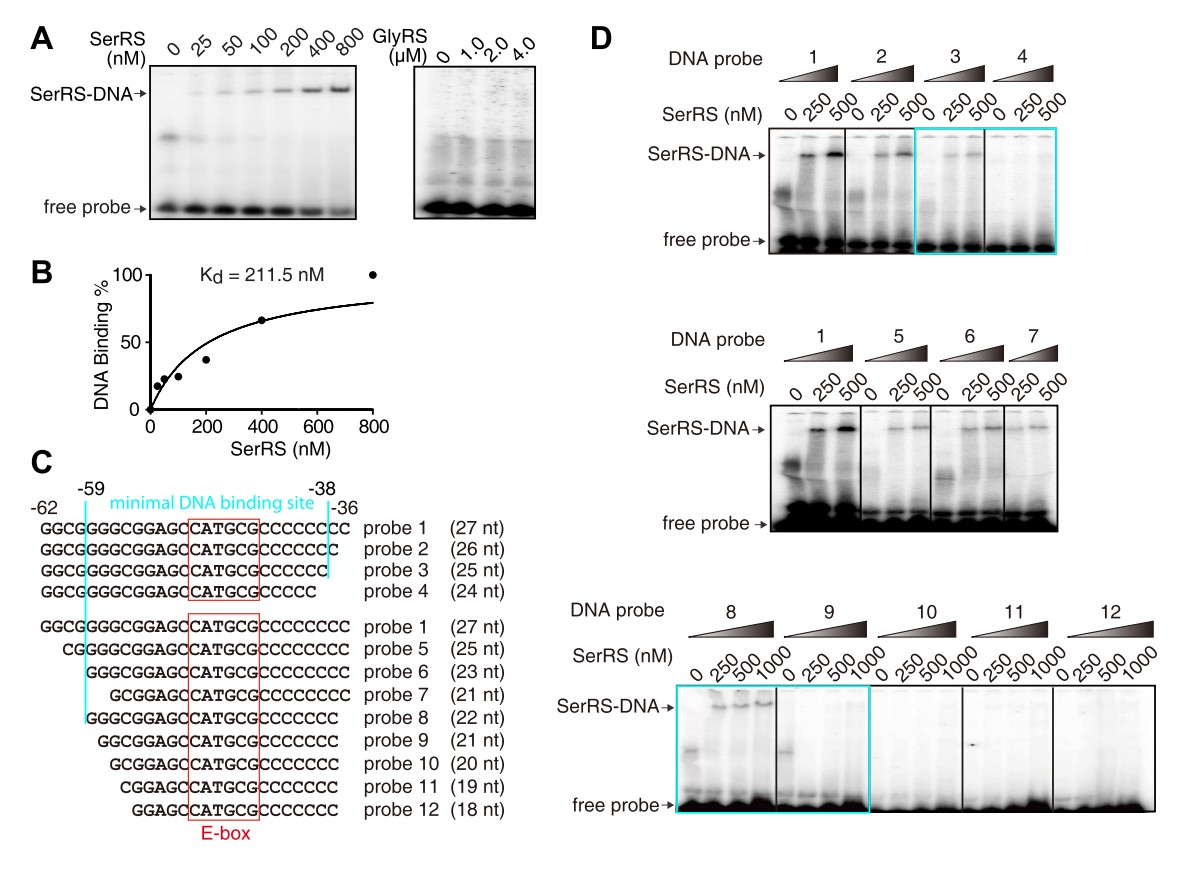

**Figure 3**. Characterization of the interaction between SerRS and DNA. (**A** and **B**) In vitro EMSA assay to determine the binding affinity between SerRS and the 27-bp DNA. The 27-bp DNA fragment containing SerRS binding site on the *VEGFA* promoter (−62 ∼ −36) were labeled by $^{32}$P at the 5' end, and then incubated with purified SerRS or GlyRS at indicated concentrations. The SerRS–DNA complex was followed by electrophoresis on native acrylamide gels. (**C** and **D**) EMSA to determine the minimal SerRS binding site on the *VEGFA* promoter. Truncations of the DNA from either end weakened the SerRS–DNA interaction. Purified recombinant SerRS protein was used at the indicated concentrations.

The following figure supplements are available for figure 3:

**Figure supplement 1**. Determination of the binding affinity between SerRS and DNA by SPR.

the nucleus. Deletion of TBD or UNE-S dramatically weakens or completely abolishes the DNA interaction (*Figure 4B*). In fact, only the intact SerRS can bind to DNA (*Figure 4B*), suggesting that multiple domains of SerRS contribute to the DNA interaction.

To further define the DNA binding sites on SerRS, we made additional deletion mutants of SerRS. Deletion of each of the two higher eukaryote-specific insertions in TBD and CD, respectively, which does not negatively impact tRNA binding (*Xu et al., 2013*), dramatically weakens the DNA interaction (*Figure 4C*). Two additional deletions—ΔV2-G14 in TBD and ΔT413-V420 in CD—also abolish the DNA binding (*Figure 4C*). Based on these results and our previously solved crystal structures of human SerRS (*Xu et al., 2012*, *2013*), we modeled the SerRS–DNA interaction. As shown in *Video 1*, motif V2-G14 and loop T413-V420, located next to each other in 3D space, bind to one end of the DNA, while insertion I in TBD (G75-N97) binds to the other end; insertion II in CD (G254-N261), as well as the UNE-S domain, which is disordered in the crystal structure of SerRS, would interact with the middle region of the DNA near the E-box.

## SerRS competes with c-Myc for binding to the *VEGFA* promoter

The overlapping DNA binding sites of SerRS and c-Myc on the *VEGFA* promoter and their opposing roles on VEGFA expression, suggest that SerRS may compete with c-Myc for DNA binding and thus inhibit c-Myc-driven VEGFA expression. It is important to note that the binding affinity of the

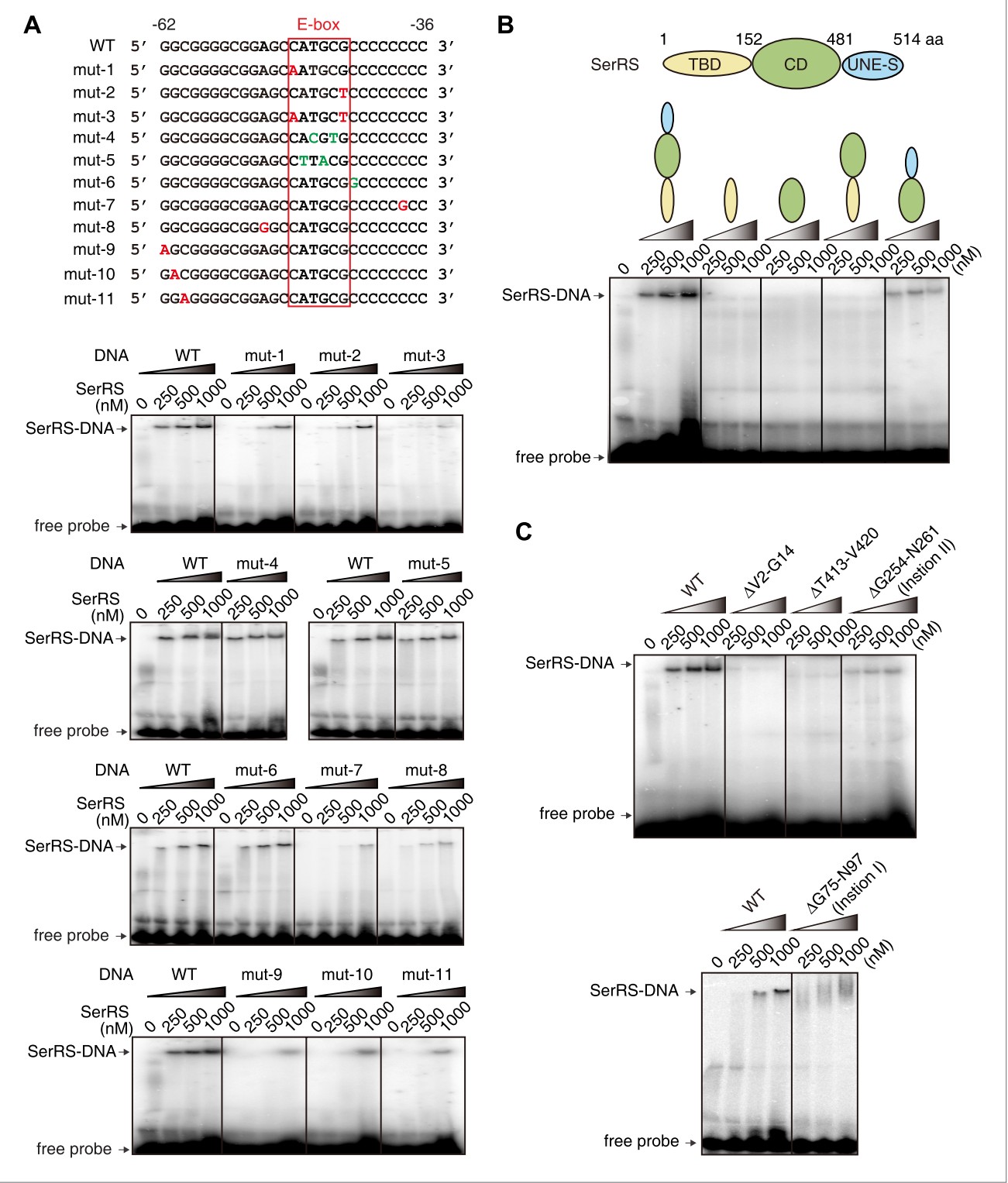

**Figure 4**. Further characterization of the interaction between SerRS and DNA. (**A**) EMSA assay to probe the DNA sequence specificity for interacting with SerRS. DNA mutations that do or do not impact SerRS binding are colored in red and green, respectively. (**B**) Domain mapping analysis and EMSA assay to reveal multiple DNA binding sites on SerRS. TBD: tRNA binding domain; CD: catalytic domain; UNE-S: C-terminal appended domain unique to vertebrates. (**C**) Deletion mutagenesis to further define *DNA binding sites on SerRS*. Deletion of either insertion I, insertion II, motif V2-G14, or loop T413-V420 greatly weakens or abolishes the DNA interaction.

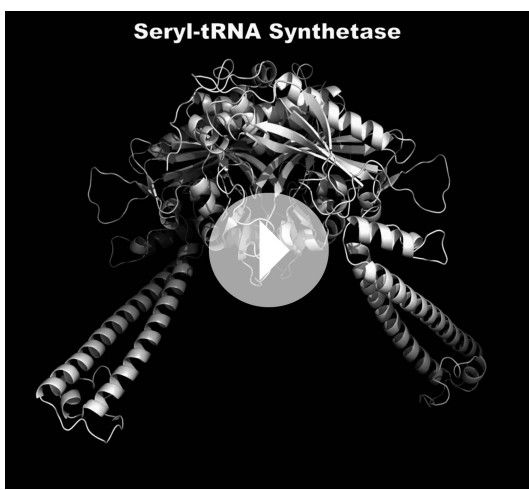

**Video 1**. Model for SerRS–DNA interaction.

SerRS–DNA interaction (*Figure 3A,B*, *Figure 3—figure supplement 1*) is comparable to that of the c-Myc/MAX–DNA interaction (90.5 ~ 229 nM) (*Hu et al., 2005*). Indeed, in HEK 293 cells and measuring by real time qRT-PCR, ectopic expression of WT (but not NLS-deleted) SerRS repressed the overexpression of VEGFA driven by c-Myc (*Figure 5A*). Consistently, our EMSA and Western blot analyses clearly showed that SerRS, at the same concentrations, could compete in vitro with the c-Myc/MAX complex for binding to the 27-bp DNA fragment from the *VEGFA* promoter (*Figure 5B*, *Figure 5—figure supplement 1*). We note that SerRS cannot compete with the MAX/MAX homodimer for binding to the DNA at comparable concentrations (*Figure 5B*, *Figure 5—figure supplement 1*), presumably because of the tight DNA binding affinity of the MAX/MAX homodimer (19.2 ~ 48.7 nM) (*Hu et al., 2005*).

We also demonstrated that SerRS competes with c-Myc for binding to the *VEGFA* promoter in whole cells. ChIP analysis showed that ectopically expressed WT, but not NLS-deleted, SerRS could compete c-Myc off of the *VEGFA* promoter in HEK 293 cells (*Figure 5C*). Consistently, knocking down endogenous SerRS expression in HUVEC cells resulted in a dramatic increase of endogenous c-Myc binding to the *VEGFA* promoter (*Figure 5D*). This increase was completely reversed when the cells were compensated with ectopically expressed WT, but not NLS-deleted, SerRS (*Figure 5D*). These results suggest that SerRS is a potent endogenous inhibitor of c-Myc for binding to the *VEGFA* promoter, and vice versa.

## 'Yin-Yang' regulation of SerRS and c-Myc in vascular development in zebrafish

Given the competition between c-Myc and SerRS for binding to the *VEGFA* promoter and the opposing activity of c-Myc and SerRS in regulating VEGFA expression, we postulated that knocking down c-Myc, although toxic on its own, may have a rescue effect towards the vasculature abnormality caused by a SerRS deficiency. This possibility was investigated in zebrafish as a vertebrate model system. As expected (*Fukui et al., 2009*), knocking down SerRS by injection of an antisense morpholino (SerRS-MO) resulted in a hyper-intersegmental vessel (ISV) branching phenotype in zebrafish (*Figure 5E*), and the phenotype was accompanied with an elevated level of Vefga expression (*Figure 5—figure supplement 2*). Specifically, out of 130 fish embryos injected with SerRS-MO, 72 (55.4%) exhibited hyper-ISV phenotype, as oppose to 2.9% (n = 4 out of 140) of fish injected with a control morpholino (control-MO). Although a small number of fish injected with SerRS-MO exhibited the opposite vascular defect (hypo-ISV phenotype), the number is not significantly different from that in the control-MO group (*Figure 5E*). Overall, without SerRS, there is an over-expansion of the vasculature.

In contrast, knocking down c-Myc by injecting an antisense morpholino against Myca (c-Myc homologue in zebrafish) showed under-developed vasculature, which is accompanied with a reduced level of Vefga expression (*Figure 5—figure supplement 2*). In particular, 41.4% (n = 24 out of 58) of Myca-MO-injected morphants exhibits hypo-ISV phenotype, as opposed to 10% (n = 14 out of 140) of control-MO-injected morphants (*Figure 5E*, *Figure 5—figure supplement 3*). Remarkably, co-injection of Myca-MO with SerRS-MO efficiently rescued both hyper-ISV (10.7%, n = 18 out of 169) and hypo-ISV (10.1%, n = 17 out of 169) defects (*Figure 5E*). Coincidently, co-injection of Myca-MO also partially reversed the high Vegfa expression level in SerRS-MO-injected zebrafish (*Figure 5—figure supplement 2*). Therefore, a counteracting effect between c-Myc and SerRS in vascular development was confirmed in a vertebrate system. These results highlight a 'Yin-Yang' regulation of SerRS and c-Myc on VEGFA expression and demonstrate that a delicate balance between them is essential for developing a functional vasculature.

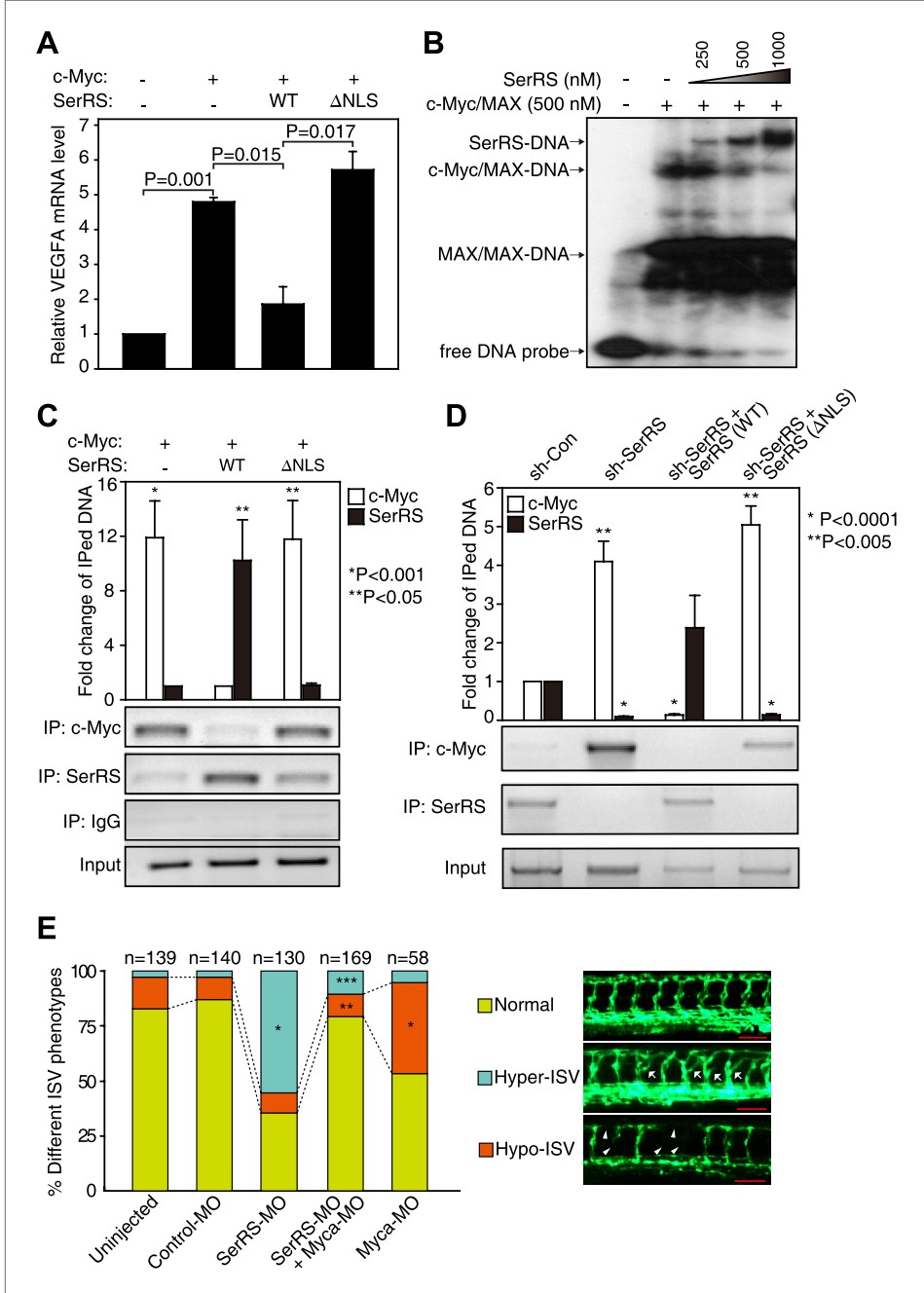

**Figure 5**. Competition between SerRS and c-Myc for DNA binding and their opposing effect in vascular development. (**A**) Competition between c-Myc and SerRS on VEGFA expression. HEK 293 cells were transfected with c-Myc alone or c-Myc with WT or ΔNLS SerRS. The mRNA levels of VEGFA were determined by RT-PCR. Values are shown as means ± SEM (n = 3). (**B**) Competition between c-Myc/MAX and SerRS for DNA binding in vitro as examined by EMSA. The 27-bp DNA was radio-labeled and incubated with purified recombinant c-Myc/MAX together with purified recombinant SerRS at indicated concentrations. The protein–DNA complexes were followed by electrophoresis on a native acrylamide gel. (**C**) Competition between ectopically expressed SerRS and c-Myc for DNA binding on the *VEGFA* promoter in HEK 293 cells as examined by ChIP. HEK 293 cells were co-transfected with plasmids expressing c-Myc and WT or ΔNLS SerRS or empty vector (−) 24 hr prior to ChIP analysis. The amounts of DNA immunoprecipitated by anti-SerRS or anti-c-Myc antibodies or by control IgG from HEK 293 cell lysates were measured by PCR using a primer set targeting the *VEGFA* promoter. The normalized results (top panel) are represented as fold change of immunoprecipitated DNA by anti-SerRS vs anti-c-Myc and are shown as means ± SEM (n = 3, *p<0.001, **p<0.05).
*Figure 5. Continued on next page*

*Figure 5. Continued*

The bottom panel shows representative gel images. (**D**) Competition between endogenously expressed SerRS and c-Myc for DNA binding on the *VEGFA* promoter in HUVECs. HUVECs were infected to express the indicated molecules 48 hr prior to ChIP analysis. The same ChIP experiment and data analysis were performed as described in (**C**). *p<0.0001, **p<0.005. (**E**) Opposing effect of SerRS and c-Myc in zebrafish vascular development and their mutual phenotypic rescue. The percentage of *Tg(Fli1a:GFP)* zebrafish embryos showing different ISV phenotypes at 3 days post fertilization after the injection of morpholinos targeting SerRS (SerRS-MO), Myca (Myca-MO), or a control morpholino (Control-MO) are illustrated. Scale bars represent 0.125 mm. *p<0.0001 vs Control-MO, **p<0.0001 vs Myca-MO, ***p<0.0001 vs SerRS-MO. Control-MO was added to SerRS-MO or Myca-MO experiments in order to maintain a constant level of total morpholinos in each experiment.

The following figure supplements are available for figure 5:

**Figure supplement 1**. SerRS and c-Myc/MAX do not simultaneously bind to the DNA.

**Figure supplement 2**. Effect of knocking down Myca or SerRS on Vegfa expression in zebrafish.

**Figure supplement 3**. Design and efficiency of the antisense morpholino against Myca.

## SerRS directly interacts with histone deacetylase SIRT2

A large scale protein–protein interaction study indicated a potential interaction between SerRS and sirtuin 2 (SIRT2) (*Ewing et al., 2007*), a NAD+-dependent histone deacetylase of the sirtuin family that regulates a broad range of processes, including transcription, metabolism, neurodegeneration, and aging (*Finkel et al., 2009*). Of the seven mammalian sirtuin isoforms, relatively little is known about SIRT2. Considering that c-Myc activates gene expression by recruiting partners harboring histone acetyltransferase activity that modifies histones and leads to open chromatin structures (*McMahon et al., 1998*, *2000*; *Amati et al., 2001*), we postulated that an interaction between SerRS and SIRT2 might reverse this process to attenuate VEGFA expression. If this were true, then by implementing opposing deacetylase/acetyltransferase activities, the 'Yin-Yang' relationship between SerRS and c-Myc would also act at the level of chromatin modification.

To test this hypothesis, we firstly performed coimmunoprecipitation to confirm the interaction. The ectopically expressed SIRT2, but not SIRT1, effectively pulled down SerRS, and vice versa (*Figure 6A*). The interaction between endogenous SerRS and SIRT2 was also confirmed by coimmunoprecipitation (*Figure 6—figure supplement 1*). Moreover, a purified GST-SerRS fusion protein successfully pulled down purified SIRT2, thus showing that the protein–protein interaction is direct (*Figure 6B*). Further mapping analysis showed that the catalytic domain (CD) of SerRS is responsible for the interaction with SIRT2 (*Figure 6B*).

To test whether SerRS can recruit SIRT2 to the *VEGFA* promoter, we knocked down the expression of SerRS in HEK 293 cells and detected the binding of SIRT2 to the *VEGFA* promoter. Knockdown of SerRS, but not GlyRS, significantly reduced the amount to SIRT2 bound to the *VEGFA* promoter (*Figure 6—figure supplement 2*), demonstrating that SerRS specifically recruits SIRT2 to the *VEGFA* promoter.

## SerRS interaction promotes SIRT2 deacetylase activity

If SIRT2 were to be recruited by SerRS to reverse histone acetylation, their interaction should not negatively affect the enzymatic activity of the deacetylase. To address this question, we first mapped the SerRS binding site on SIRT2 by coimmunoprecipitation (*Figure 6C*). SIRT2 is a 389-aa protein, whose crystal structure has been solved (*Finnin et al., 2001*). The catalytic core of SIRT2 is flanked by ~60 aa and ~50 aa on the N- and C-terminal ends, respectively. Through a series of truncations from each end, we determined that both N- and C-terminal regions, outside the catalytic core of SIRT2, interact with SerRS (*Figure 6C*). Particularly, residues within the regions of G52-D60 and W337-S356 are critical, as the loss of either region abolished the interaction of SIRT2 with SerRS. Interestingly, both regions are located on the opposite side of the substrate-binding pocket (*Figure 6D*), suggesting that bound SerRS should not interfere with the deacetylase activity of SIRT2. To confirm the postulation, we directly assessed the effect of SerRS on the in vitro deacetylase activity of SIRT2. Remarkably, at an equal molar ratio (1:1), SerRS did not inhibit but rather promoted the deacetylase activity of SIRT2

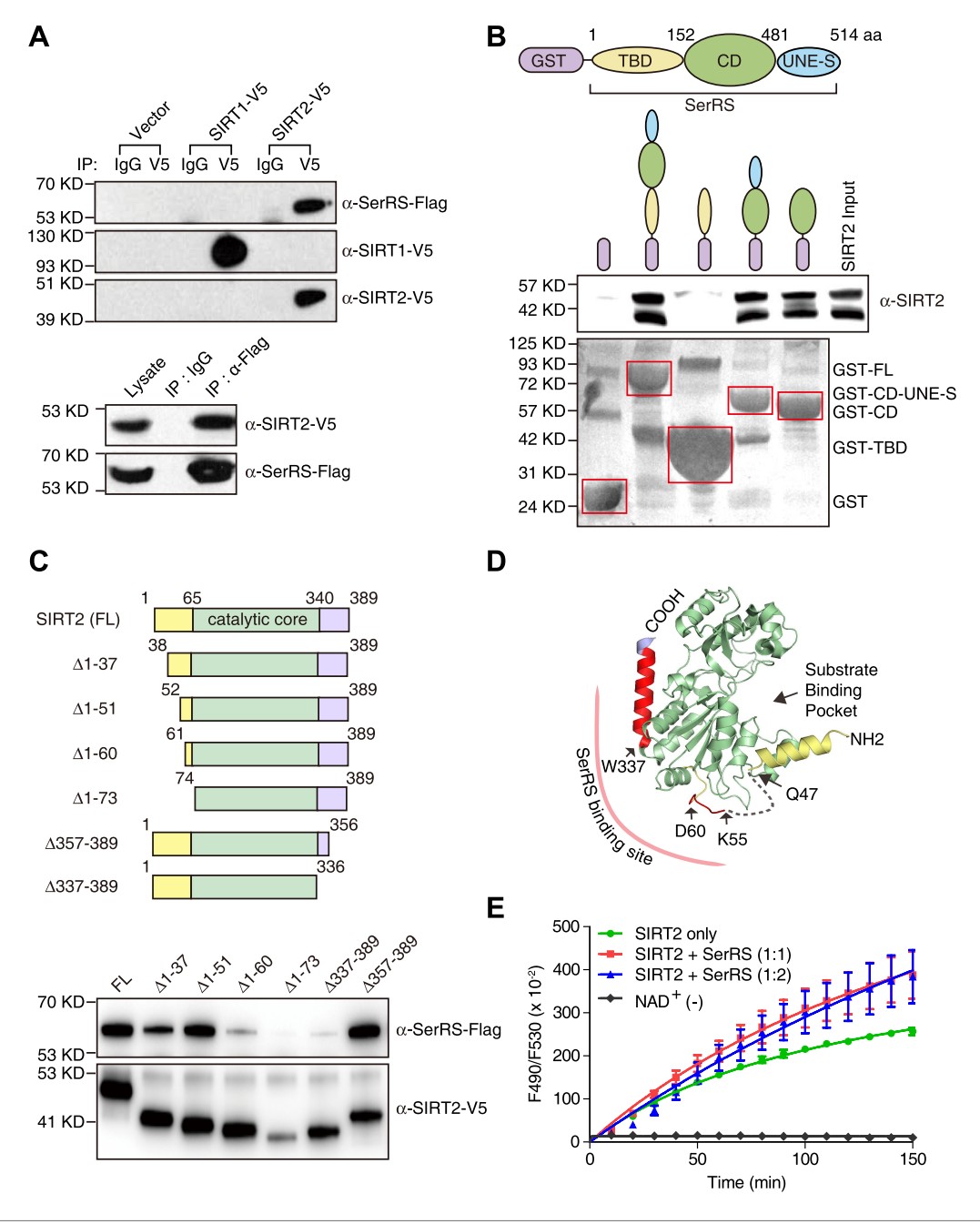

**Figure 6**. Demonstration and characterization of SerRS/SIRT2 interaction. (**A**) SerRS specifically interacts with SIRT2 but not SIRT1. HEK 293 cells were co-transfected with plasmids expressing Flag-tagged SerRS and V5-tagged SIRT1 or SIRT2. Cell lysate was immunoprecipitated with anti-V5 (top panel), anti-Flag (bottom panel) antibodies or control IgG. The experiment was followed by Western blot analysis to detect the interaction between SerRS and SIRT1/SIRT2 using anti-Flag and anti-V5 antibodies. (**B**) GST-pull down assay to show that SerRS/SIRT2 interaction is direct and that the interaction is mediated by the catalytic domain of SerRS. Full-length SerRS or its domain fragments were fused with GST at N-termini to pull down purified His-tagged SIRT2. SIRT2 was detected by Western blot analysis using anti-His$_6$ antibody, and the GST fusion proteins attached on the Glutathione-Sepharose beads were analyzed using ponceau S staining. TBD: tRNA-binding domain; CD: catalytic domain; UNE-S: C-terminal appended domain. (**C**) Mapping study to identify the SerRS binding sites on SIRT2. V5-tagged full-length SIRT2 or its truncated fragments was co-transfected with Flag-tagged SerRS into HEK 293 cells. SIRT2 proteins were immunoprecipitated with anti-V5 antibody and the SIRT2-bound SerRS proteins were detected by Western blot using anti-Flag antibody.
*Figure 6. Continued on next page*

*Figure 6. Continued*

(**D**) Illustration of the SerRS binding sites on the structure of SIRT2. Two SerRS binding sites (Gly52-Asp60, Trp337-Ser356) are highlighted in red. The catalytic domain of SIRT2 is in green, while the partially disordered N- and C-terminal regions are in yellow and purple, respectively. The gray dash line represents a disordered internal region. (**E**) Effect of SerRS on SIRT2 deacetylation activity. Recombinant human SIRT2 (1 μM) were incubated with purified SerRS (concentration measured as monomer) at the indicated ratios. The deacetylase activities of SIRT2 were measured by using a substrate peptide with one end coupled to a fluorophore and the other end to a quencher. An internal acetylated lysine residue serves as the substrate of SIRT2, and the deacetylation allows the peptide to be cleaved by a lysylendopeptidase to release the fluorophore from the quencher to emit fluorescence. Therefore, the SIRT2 acitivity was measured by monitoring the fluororescence intensity (excitation at 490 nm and emission at 530 nm). A reaction without NAD$^+$ (NAD$^+$ [−]) was performed as a negative control.

The following figure supplements are available for figure 6:

**Figure supplement 1**. Endogenous interaction between SerRS and SIRT2.

**Figure supplement 2**. SerRS recruits SIRT2 to the *VEGFA* promoter.

(*Figure 6E*). Because increasing the SerRS concentration to 1:2 ratio (SIRT2: SerRS) did not provide any additional enhancement (but rather a small decline at early time points), the enhancement effect is likely to result from the specific interaction between SerRS and SIRT2, at a location distal to the active site.

## SerRS recruits SIRT2 to epigenetically silence VEGFA expression

Next, we tested whether SerRS can recruit SIRT2 to modify the histone modification on the *VEGFA* promoter. Because the evolutionarily conserved deacetylase activity of SIRT2 has a strong preference for K16 of histone H4 (*Vaquero et al., 2006*), we performed ChIP analyses using antibodies against acetylated H4K16 (H4K16Ac). Remarkably, overexpression of SerRS, but not of GlyRS, reduced the level of H4K16Ac on the *VEGFA* promoter (*Figure 7A*). Consistently, knocking down endogenous SerRS, but not endogenous GlyRS, had the opposite effect and significantly increased H4K16Ac on the *VEGFA* promoter (*Figure 7B*, *Figure 7—figure supplement 1A*). The increase was reversed when the cells were compensated with WT, but not NLS-deleted, SerRS (*Figure 7B*). These results demonstrated that nuclear SerRS acts to decrease the amount of acetylated H4 on the *VEGFA* promoter. Given the interaction between SerRS and SIRT2, this effect on H4 acetylation is presumably through engagement of SIRT2 by SerRS.

To confirm that SIRT2 is a necessary cofactor for SerRS to repress VEGFA expression, we disrupted SIRT2 by RNAi. Indeed, knocking down the expression of SIRT2, but not of SIRT1, completely reversed the transcriptional repression activity of SerRS on VEGFA expression (*Figure 7C*, *Figure 7—figure supplement 1B*). Consistently, inhibiting SIRT2 activity by AGK2, a SIRT2-specific inhibitor (*Outeiro et al., 2007*), also completely knocked out the transcriptional repression activity of SerRS, while EX-527, a SIRT1 inhibitor (*Solomon et al., 2006*), had little effect (*Figure 7D*). Therefore, we have demonstrated that, by recruiting SIRT2 vs a histone acetyltransferase, the 'Yin-Yang' relationship between SerRS and c-Myc also acts at the level of chromatin modification.

## SIRT2 inhibits VEGFA expression and vasculogenesis in zebrafish

While SIRT1 has been shown to promote angiogenesis (*Potente et al., 2007*), the role of SIRT2 in vascular development has not been clear. The critical role of SIRT2 in the mechanism of SerRS to inhibit VEGFA expression and vascular expansion suggests that SIRT2 could be associated with an anti-angiogenic function, and that knocking down SIRT2 may mimic the vasculature abnormality phenotype caused by a SerRS knockdown. We tested this hypothesis in the zebrafish system. Consistent with the previous report showing a VEGFA-independent pro-angiogenic role for SIRT1 in vascular development (*Potente et al., 2007*), injection of an antisense morpholino against Sirt1 (SIRT1 homologue in zebrafish) generated the hypo-ISV phenotype (*Figure 7E*) and had no significant effect on Vegfa expression (*Figure 7F*). In contrast, injection of a *Sirt2-MO* resulted in the same hyper-ISV phenotype (42.1%, n = 75 out 178) as with the injection of SerRS-MO (46.7%, n = 119 out of 255) (*Figure 7E*, *Figure 7—figure supplement 2A,B*). Furthermore, the hyper-ISV phenotype, in both cases, was accompanied by a significantly elevated level of the *vegfa* transcript (*Figure 7F*). Therefore, our

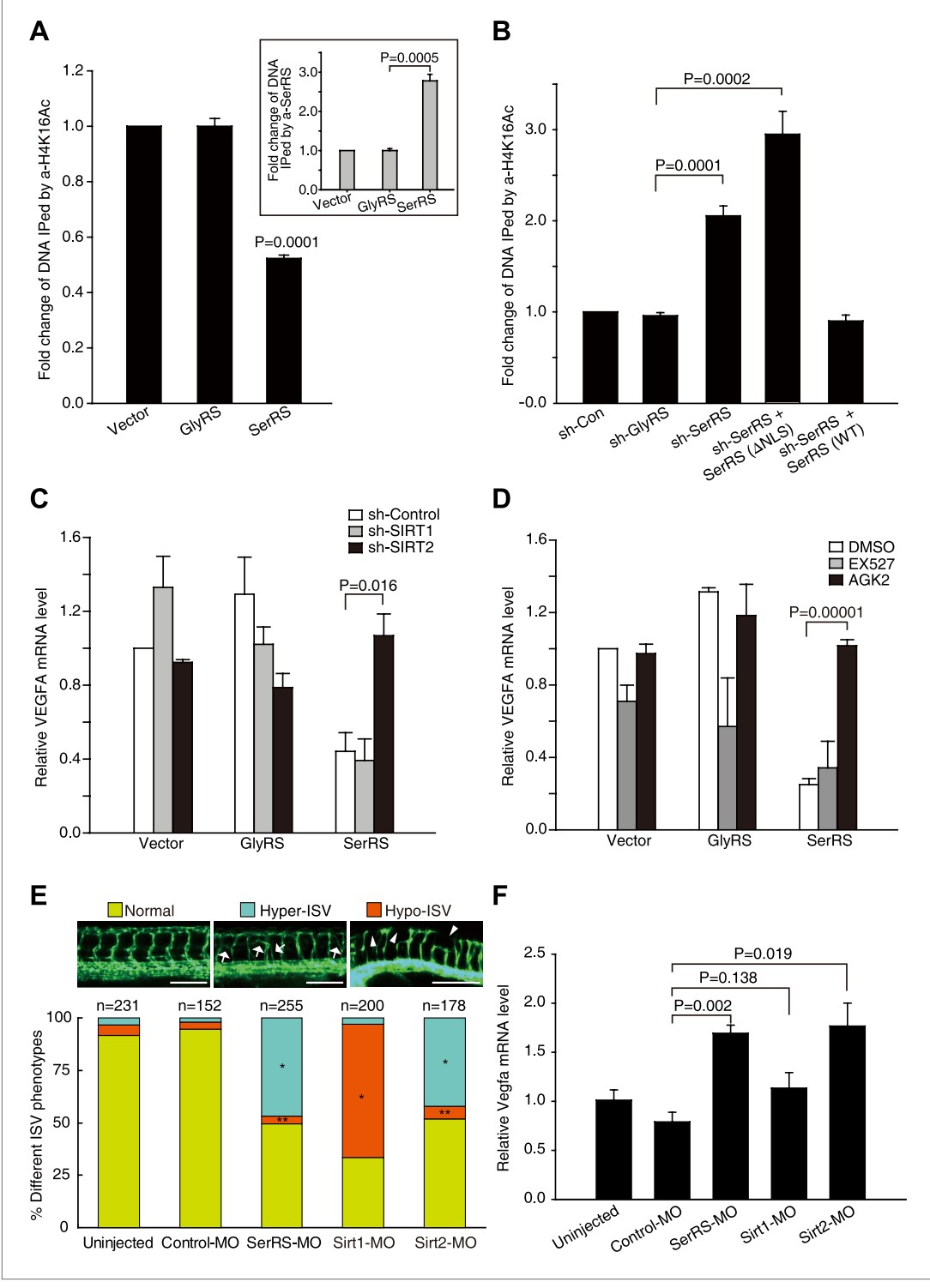

**Figure 7**. SerRS recruits SIRT2 to epigenetically silence VEGFA expression. (**A**) ChIP assay to show that overexpression of SerRS reduces histone H4 acetylation level on the *VEGFA* promoter. HEK 293 cells were transfected with plasmids expressing SerRS, GlyRS or empty vector. The cell lysates were subjected to local ChIP analysis using anti-H4K16Ac (acetylated H4 at K16), anti-H4 (total), or anti-SerRS antibodies and a primer set targeting the *VEGFA* promoter. The amounts of DNA immunoprecipitated by anti-H4K16Ac antibody were normalized to those by anti-H4 antibody prior to fold change calculation. Inset: the normalized amounts of DNA immunoprecipitated by anti-SerRS. All data were shown as means ± SEM (n = 3). (**B**) ChIP assay to show that knock down of SerRS expression or exclusion of
*Figure 7. Continued on next page*

*Figure 7. Continued*

SerRS from the nucleus increases histone H4 acetylation level on the *VEGFA* promoter. HEK 293 cells were transfected with plasmids expressing the indicated molecules and subjected to local ChIP analysis as described above. As a control, GlyRS expression was knocked down but had no effect on H4 acetylation. (**C**) Effect of SIRT2 expression on the transcriptional repressor activity of SerRS as measured by VEGFA expression. HEK 293 cells were co-transfected with plasmids expressing shRNAs targeting SIRT1, SIRT2 or control shRNA and plasmids expressing SerRS, GlyRS or empty vector for 36 hr. The VEGFA expression levels were determined by using real-time RT-qPCR and are shown as means ± SEM (n = 3). (**D**) Effect of SIRT2-specific inhibitor on the transcriptional repressor activity of SerRS as measured by VEGFA expression. HEK 293 cells were transfected with plasmids expressing SerRS, GlyRS or empty vector. SIRT2-specific inhibitor AGK2 (10 μM, final concentration) or SIRT1-specific inhibitor EX-527 (1 μm, final concentration) or solvent alone (DMSO) was added to the cell culture media 2 hr post-transfection. VEGFA expression levels were measured 24 hr post-transfection by using real-time RT-qPCR and are shown as means ± SEM (n = 3). (**E**) Functional correlation between SerRS and SIRT2 in zebrafish. The percentage of *Tg(Fli1a:EGFP)* zebrafish embryos showing different ISV phenotypes at 3 days post fertilization after the injection of morpholinos targeting SerRS (SerRS-MO), Sirt1 (Sirt1-MO), Sirt2 (Sirt2-MO), or a control morpholino (Control-MO) are illustrated. Scale bars represent 0.25 mm. *p<0.0001 vs Control-MO, **p>0.1 vs Control-MO. (**F**) The effects of knocking down SerRS, Sirt2, or Sirt1 in zebrafish on Vegfa expression were examined by real-time RT-qPCR at 1 day post fertilization after injection of morpholinos as indicated. Data are shown as means ± SEM (n = 10–15).

The following figure supplements are available for figure 7:

**Figure supplement 1**. Knock-down efficiencies of shRNAs targeting GlyRS, SerRS, SIRT1, and SIRT2.

**Figure supplement 2**. Design and efficiency of the antisense morpholino against Sirt2.

---

studies have revealed an anti-angiogenic role of SIRT2, which should be, at least in part, dependent on attenuating VEGFA expression through its interaction with SerRS.

## Discussion

Through in vitro, cell-based and animal experiments, we established that the essential role of SerRS in vascular development arises from its novel activities as a transcriptional repressor of VEGFA. There are two different aspects of this activity: first, SerRS directly binds to the *VEGFA* promoter; second, DNA-bound SerRS recruits the SIRT2 histone deacetylase to condense the chromatin at the *VEGFA* promoter, and thereby shut down the gene transcription. Importantly, in each aspect, these actions of SerRS directly compete with and thwart that of the VEGFA-promoting actions of c-Myc. While the opposing regulation of SerRS and c-Myc is applied on VEGFA expression, it is manifested at the organism level with respect to vascular development, making SerRS and c-Myc as a pair of 'Yin-Yang' regulators for proper development of a functional vasculature (*Figure 8*).

With the same bHLHZ domain as in c-Myc and MAX, Mad family proteins (comprised of Mad1, Mxi1, Mad3, and Mad4) can compete with c-Myc for binding to MAX and also recruit histone deacetylases to reverse the action of c-Myc-bound acetyltransferase to shut down the expression of c-Myc target genes (*Grandori et al., 2000*). Therefore, Mad proteins are generally considered as antagonists of c-Myc, especially with regard to the role of c-Myc in tumorigenesis (*Zhou and Hurlin, 2001*). However, the role of Mad proteins in vascular development appears to be non-essential. Disruption of members of the Mad family in mice does not exhibit any vascular phenotype (*Foley et al., 1998*; *Schreiber-Agus et al., 1998*). In addition, the temporal expression pattern varies between c-Myc and the Mad family proteins, with c-Myc being expressed during development, while expression of Mad proteins is mainly induced during terminal differentiation (*Queva et al., 1998*). Thus, the Mad proteins are largely silent during the time that the vasculature is being established, which could explain their non-essential role in vascular development. In comparison, SerRS' ubiquitous expression as an essential tRNA synthetase could match better with that of c-Myc to provide the counterbalance.

From an evolutionary perspective, a closed circulatory system with vasculature network is one of the hallmarks of vertebrates. Although c-Myc plays a key role in vascular development, the gene first appeared in *Drosphila*, an invertebrate with a primitive 'open' circulatory system. The appearance of the MAX and Mad family genes is even earlier, and the genes were first identified in roundworms such as *C. elegans* (*Atchley and Fitch, 1995*; *Prendergast, 1999*). In contrast, although SerRS is considered as one of the most ancient proteins, its UNE-S domain, which harbors the NLS signal to endow

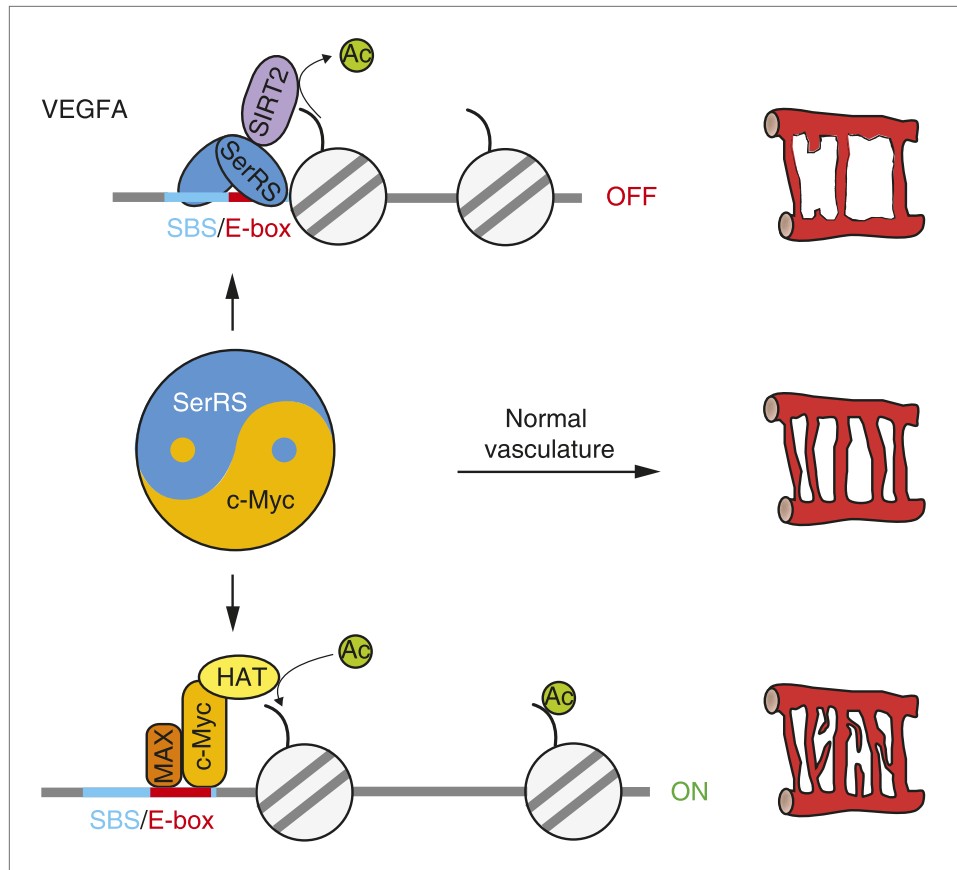

**Figure 8**. The 'Yin-Yang' relationship of SerRS and c-Myc in vascular development. Nuclear SerRS binds to the *VEGFA* promoter at the identified SerRS binding site (SBS) and recruits the SIRT2 histone deacetylase to condense the local chromatin to shut down VEGFA expression. These tandem actions of SerRS symmetrically offset the VEGFA-promoting actions of c-Myc to maintain a delicate balance for the development of a functional vasculature.

SerRS with its novel transcription repressor activity, only appeared in vertebrates. Thus, it seems that an 'old' tRNA synthetase evolved to function as a 'new' and essential antagonist against c-Myc for proper development of the advanced closed circulatory system of vertebrates.

A critical component of the c-Myc-antagonizing role of SerRS is the recruitment of SIRT2. In fact, a complete reversal of the inhibitory effect of SerRS on VEGFA expression was observed when SIRT2 was knocked out by RNAi or inhibited by EX527 (*Figure 7C,D*). This observation indicates that the transcriptional repressor role of SerRS is ultimately through SIRT2 and that the direct blocking of c-Myc from the promoter by SerRS may have lesser significance for downregulating gene expression. Thus, an overlapping DNA binding site may not be a prerequisite for SerRS to antagonize c-Myc, as long as both SerRS and c-Myc bind to the same promoter. Considering the large number of genes that are regulated by c-Myc, one would not be surprised to find additional genes to be transcriptionally repressed by SerRS, presumably also through its collaboration with SIRT2.

It is worth noting that SIRT2 has been identified as a tumor suppressor (*Hiratsuka et al., 2003*; *Lennerz et al., 2005*), while c-Myc is a prominent oncogene that promotes tumor cell proliferation and tumor vascularization (*Baudino et al., 2002*). We speculate that SerRS also functions as a tumor suppressor by collaborating with SIRT2 to antagonize c-Myc. Interestingly, human *SARS* is located on the short arm of chromosome 1 (i.e., 1p13.3), which is frequently affected by rearrangements or allelic loss in a variety of human malignancies (*Morgan et al., 1985*; *Mitchell and Santibanez-Koref, 1990*; *Mathew et al., 1994*; *Munn et al., 1995*; *Nagai et al., 1995*; *Xu et al., 2001*; *Caramazza et al., 2009*). The frequent disruption of this chromosomal locus in human malignancies suggests the presence of tumor suppressor genes which, when perturbed, lead to increased cancer susceptibility. Because SerRS is also essential for survival through its function in protein synthesis, if it is a factor in

any of these malignancies, then its function in translation would need to be preserved. This could be achieved by disruption of the UNE-S domain, which does not affect aminoacylation, but is essential for SerRS to antagonize c-Myc.

## Materials and methods

### Antibodies

Custom-made rabbit anti-human SerRS antibody was raised against purified human recombinant SerRS and affinity-purified. Monoclonal anti-SerRS antibody for coimmunoprecipitation was purchased from Abnova (Taipei, Taiwan). Anti-c-Myc, anti-SIRT2, anti-SIRT1, and anti-α-tubulin antibodies were purchased from Cell Signaling (Danvers, MA, USA). Anti-V5 and anti-GlyRS antibodies were purchased from Invitrogen (Grand Island, NY, USA) and Abnova (Walnut, CA, USA), respectively. Antibodies against histone H4 and acetylated H4 at Lys 16 (H4K16Ac) were purchased from Active Motif (Carlsbad, CA, USA). Anti-SIRT2 antibody for chromatin immunoprecipitation was purchased from Thermo Fisher (Rockford, IL, USA).

### Protein expression and purification

For overexpressions in mammalian cells, human full-length, and NLS-deleted SerRS genes were cloned into the pFlag-CMV-2 vector (Sigma-Aldrich, St. Louis, MD, USA), and human c-Myc, SIRT1, and SIRT2 genes into the pCDNA6-V5/His-C vector (Life Technologies, Grand Island, NY, USA). For recombinant protein purification, human SerRS, c-Myc, MAX, and SIRT2 genes were subcloned into pET-20b(+) vector (Novagen, Darmstadt, Germany) to express with a C-terminal his-tag in *Escherichia coli*. The SerRS proteins were purified in tandem by Ni-NTA affinity (Qiagen, Valencia, CA, USA), HiTrap Heparin High Performance (GE Healthcare, Pittsburgh, PA, USA), and HiLoad 16/600 Superdex 200 pg (GE Healthcare) columns. The GST-tagged SerRS constructs were subcloned into the pGEX-6P-1 vector (GE Healthcare) for expression in *E. coli*, and the proteins were affinity-purified using Glutathione-Sepharose 4B beads (GE Healthcare). The purities of the recombinant proteins were assessed by Coomassie blue staining following 4–12% Mini Gel (Life Technologies, Grand Island, NY, USA) electrophoresis. Protein concentrations were determined using Bradford protein assay (BioRad, Hercules, CA, USA).

### Cell culture and shRNAs

HEK 293 cells were cultured in DMEM supplemented with 10% fetal calf serum (FCS) and transfected with Lipofectamine 2000 (Life Technologies, Grand Island, NY, USA). HUVEC cell were cultured in EGM complete medium (Lonza, Allendale, NJ, USA) supplemented with 8% FCS in gelatin-coated dishes and transfected using lentivirus. DNA expressing a short-hairpin RNA (shRNA) designed against human SerRS (5′-GGCATAGGGACCCATCATTGA-3′), GlyRS (5′-GCATGGAGTATCTCACAAAGT-3′), SIRT1 (5′-GAAGTTGACCTCCTCATTGTT-3′) (*Guarani et al., 2011*), or SIRT2 (5′-GGACAACAGAGAGGGAGAAAC-3′) gene was inserted into the pLentiLox-hH1 plasmid, modified from the pLentiLox 3.7 plasmid to contain a H1 promoter (between Xba I and Xho I sites) to drive the shRNA expression. To compensate for the loss of endogenous SerRS expression, the coding region for GFP in the pLentiLox-hH1 plasmid was replaced with NLS-deleted or WT (as control) SerRS coding sequences. All designed shRNAs target sequences within the open reading frame except for the SerRS shRNA, which targets the 3′ untranslated region in ordered to selectively knockdown the endogenous gene but not the exogenous genes. The recombinant lentiviruses were produced in packaging 293 cells by cotransfecting the pLentiLox-hH1 plasmid with two helper packaging plasmids Δ8.9 and VSVG and subsequently concentrated by centrifugation at 50,000×*g* for 3 hr.

### Quantitative RT-PCR and statistical analysis

Total RNA was isolated from cells by TRIzol Reagent (Life Technologies, Grand Island, NY, USA). One milligram of the total RNA from each sample was reversely transcribed to cDNA by SuperScript II reverse transcriptase (Life Technologies, Grand Island, NY, USA). All real-time PCR reactions were performed using the StepOnePlus Real-Time PCR system (Applied Biosystems, Grand Island, NY, USA) with SYBR Select Master Mix (Applied Biosystems, Grand Island, NY, USA). The primer pairs for the PCR reactions were: 5′-GAGGGCAGAATCATCACGAAG-3′ and 5′-TGTGCTGTAGGAAGCTCATCTCTC-3′ for human *VEGFA*; 5′-CGTCACCAACTGGGACGA-3′ and

5'-ATGGGGGAGGGCATACC-3' for human *β-ACTIN*; 5'-GGCTCTCCTCCATCTGTCTGC-3' and 5'-CAGTGGTTTTCTTTCTTTGCTTTG-3' for zebrafish *vegfa* ; 5'-TCACCACCACAGCCGAAAGAG-3' and 5'-GTCAGCAATGCCAGGGTACAT-3' for zebrafish *β-actin*. The PCR reaction program started at 95°C for 10 min, followed by 45 cycles of 95°C for 20 s and 60°C for 1 min. Each experiment was carried out in triplicate. The *VEGFA* gene expression was normalized to that of *β-ACTIN*. Statistical analyses were performed with the software SigmaPlot (version 10.0). Student's *t* test was used to analyze the changes between different groups.

## Endothelial cell tube formation assay

48 hr before the tube formation assay, HUVEC cells were infected with lentiviruses that produce different shRNAs as indicated. Pre-thawed matrigel basement membrane matrix (0.15 ml) (BD Biosciences, San Jose, CA, USA) was transferred to 48-well plates and incubated at 37°C for 30 min to form a thin layer of gel. The infected HUVEC cells ($2 \times 10^4$) were seeded on the gel and then cultured in EBM Basal Medium (without FCS) at 37°C and 5% $CO_2$ for 24 hr to form tubes. Images of the endothelia cell tubular network were taken with a Leica DC350F CCD camera attached to an inverted Leica DMIL microscope. The length of the tubes was measured by ImageJ software.

## Chromatin immunoprecipitation (ChIP)

Cells were fixed with formaldehyde (1% final concentration) for 10 min at room temperature. The reaction was stopped by adding 125 mM of glycine. ChIP assays were performed according to the protocol of ChIP-IT Express Enzymatic kit (Active Motif). After three washes, ChIPed DNA was analyzed on the StepOnePlus Real-Time PCR system using SYBR Select Master Mix (Applied Biosystems). A primer set (5'-GGGCGGATGGGTAATTTTCA-3' and 5'-CTGCGGACGCCCAGTGAA-3') targeting the *VEGFA* gene near and upstream of the transcriptional start site was used. Nine additional primer sets for scanning the *VEGFA* promoter from −4 kb to +4 kb were described previously (*Kim et al., 2007*).

## Luciferase reporter assay

The luciferase activity was determined by using the Dual-Luciferase Reporter assay system (Promega, Madison, WI, USA). The promoter regions (−1262 ~ +46; −762 ~ +46; −262 ~ +46) of *VEGFA* gene were PCR amplified and cloned into the pGL4.11[luc2P] vector (between Kpn I and Xho I sites) to create the pGL4-VEGFA firefly luciferase reporter plasmids. After 16 hr of incubation in 12-well plates, HEK 293 cells were transiently transfected with 100 ng of pGL4-VEGFA reporter plasmid and 500 ng of pFlag-SerRS, or pFlag-GlyRS or pFlag-CMV-2 empty vector as control. A Renilla luciferase control reporter plasmid pRL-SV40 (50 ng) was co-transfected for normalizing the transfection efficiency among different experiments.

## DNase I footprinting assay

The DNA of the *VEGFA* promoter from −262 to +46 bp was released from the pGL4-VEGFA plasmid by Kpn I and Xho I digestion. After purification by agarose gel electrophoresis, the 3' end was radiolabeled using a standard Klenow fragment fill-in reaction with [α-$^{32}$P]-dATP. The labeled DNA fragment was incubated with recombinant SerRS, c-Myc and MAX, or GlyRS in 20 µl binding buffer (20 mM HEPES pH 7.9, 120 mM KCl, 8 mM $MgCl_2$, 0.2 mM EDTA, 0.5 mM DTT, 0.2 mg ml$^{-1}$ bovine serum albumin [BSA], 10 µg ml$^{-1}$ poly [dG-dC], and 5% glycerol) for 1 hr at room temperature. DNase I (New England Biolabs, Ipswich, MA, USA) was then added to the mixture at a 2.5 U ml$^{-1}$ final concentration and incubated for additional 40 min at room temperature. The reaction was stopped by adding 200 µl stop solution (20 mM Tris–HCl pH 7.5, 0.1 M NaCl, 1% [wt/vol] SDS, 5 mM EDTA, and 50 µg ml$^{-1}$ protease K) to incubate for 30 min at 45°C. After extraction with phenol-chloroform and precipitation with ethanol, DNA fragments were resuspended in 80% formamide in 1x TE buffer and then denatured for 5 min at 95°C before separation by electrophoresis using 8% urea-polyacrylamide sequencing gels. Gels were dried and examined by autoradiography.

## Electrophoretic mobility shift assay (EMSA)

The 27-bp DNA oligonucleotide corresponding to SerRS binding site on the *VEGFA* promoter and mutants were synthesized, annealed, and [$^{32}$P]-labeled at the 5' end by T4 DNA kinase (New England Biolabs) before purification using a sephadex G-25 spin column (GE Healthcare). The labeled oligonucleotides (0.08 pmol) were incubated with recombinant SerRS at indicated concentrations in binding buffer (20 mM Tris-HCl, pH 8.0, 60 mM KCl, 5 mM $MgCl_2$, 0.1 mg ml$^{-1}$ BSA, 10 ng µl$^{-1}$ poly (dG-dC),

1 mM DTT ) for 30 min at room temperature. The samples were loaded to 5% native polyacrylamide gel (17.5 cm in length) and underwent electrophoresis at 250 V in running buffer (25 mM Tris, pH 8.3, 190 mM glycine). Afterwards, the gel was dried and examined by autoradiography.

## Coimmunoprecipitation assays and Western blot analysis

HEK 293 cells were resuspended on ice with lysis buffer (20 mM Tris-HCl [pH 7.5], 150 mM NaCl, 1 mM of EDTA, 1 mM EGTA, 1% Triton X-100, 2.5 mM sodium pyrophosphate, 1 mM beta-glycerophosphate, 1 mM Na$_3$VO$_4$, and protease inhibitor cocktail). Supernatants were incubated with indicated antibodies and protein-G-conjugated agarose beads (Invitrogen) for at least 2 hr. The beads were washed five times with wash buffer (same as the lysis buffer, except that Triton X-100 was reduced from 1% to 0.1%) and then subjected to SDS-PAGE and Western blotting analysis with indicated antibodies.

## In vitro pull-down assays

GST pull-down assays were performed in the buffer containing 20 mM of HEPES (pH 7.9), 150 mM of NaCl, 0.5 mM of EDTA, 10% glycerol, 0.1% Triton X-100 and 1 mM of DTT. Equal amounts of GST or GST-SerRS fusion proteins were incubated with recombinant SIRT2 for two hours and pulled down by Glutathione-Sepharose 4B beads (GE Healthcare).

## SIRT2 deacetylase activity assay

The deacetylase activity of SIRT2 was measured by using CycLex SIRT2 Deacetylase Fluorometric Assay Kit (CycLex, Nagano, Japan). The reaction buffer contains 50 mM Tris-HCl (pH 8.8), 0.5 mM DTT, 0.25 mAU/ml Lysylendopeptidase, 1 µM Trichostatin A, 0.8 mM NAD+, 20 µM Fluoro-Substrate peptide, 1 µM recombinant SIRT2, and recombinant SerRS at different ratio indicated in *Figure 6E*. The reactions were performed at room temperature. The fluorescence intensities at 10-min intervals were read using FluoroMax-3 (Jobin Yvon Inc, Edison, NJ, USA) with excitation at 490 ± 10 nm and emission at 530 ± 10 nm.

## Surface plasmon resonance analysis

Biotin-labeled double-stranded DNA oligonucleotides corresponding to SerRS binding site on the *VEGFA* promoter (5'-GGCGGG GCGGAGCCATGCGCCCCCCCCTTTATA-biotin-3' and 5'-AAAGGGGGGGGGCGCATGGCTCCGCCCCGCC-3') were synthesized, annealed, and purified by electrophoresis on 8% native acrylamide gel. Binding kinetics was analyzed using a Biacore 3000 instrument (Biacore, Inc., Piscataway, NJ, USA). The DNA was immobilized through biotin–streptavidin interaction on a SA sensor chip, and the interaction reached to 300 Response Unit. A flow cell without immobilized DNA was used as a blank reference control. The immobilized DNA was stable over the course of the experiment. Baseline drift was less than 5 RU/h after the chip was washed with HBS-EP buffer (0.01 M HEPES pH 7.4, 0.15 M NaCl, and 0.005% surfactant P-20 [vol/vol]) at 10 µl min⁻¹ for 18 hr. SerRS proteins were injected using the KINJECT procedure for 300 s at 6.25, 12.5, 25, 50, 100, 200, 400, 800, and 1600 nM concentrations in HBS-EP buffer at 30 µl min⁻¹. Dissociation was monitored by flowing HBS-EP buffer for 480 s at 30 µl min⁻¹. The sensor chip was regenerated by a 60-s injection of 0.1% SDS, 10 mM NaOH to restore the original resonance signal of the surface. The injections were duplicated for each ligand concentration and were performed in random orders with buffer blanks injected periodically for double referencing. The variation between the replicates was less than 1%. Corrected response data were fitted with BiaevalTM 3.1 software and the apparent kinetic constants were calculated using data from the early parts of the association and dissociation phases. The fit was satisfactory for a simple 1:1 binding model.

## In vivo studies in zebrafish

Transgenic *Tg* (*Fli1a: EGFP*) fish were maintained at 28.5°C under continuous water flow and filtration with automatic control for a 14:10 hr light/dark cycle. The night before injection, male and female fish were placed in a 1-L tank containing fish mating cage with an inner mesh and divider. Zebrafish embryos were obtained from natural spawning by removing the divider and stimulating with light. The embryos were kept at 28.5°C before and after microinjection. The antisense morpholinos (MOs) targeting SerRS or other genes were injected into the yolk of 1- to 2-cell stage embryos at the dosage of 4 ~ 5 ng per embryo. The designs of SerRS-MO (5'-AGGAGAATGTGAACAAACCTGACAC-3') (*Fukui et al., 2009*) and of Sirt1-MO (5'- TATTTTCGCCGTCCGCCATCTTCGC-3') have been described previously (*Potente et al., 2007*). The Myca-MO (5'-CATTTTGACACTTGAGGAAGGAGAT-3') and

Sirt2-MO (5'-CATCTGAGCAGAAACTCACATTTGC-3') were designed de novo for this study. All MOs including a standard control MO (5'-CCTCTTACC TCAGTTACAATTTATA-3') were purchased from Gene Tools, LLC (Philomath, OR, USA). After injection, embryos were incubated in E3 embryo medium supplemented with 0.003% 1-phenyl-2-thiourea (PTU) at 28.5°C to prevent pigment formation. Embryos were anesthetized with 0.168 mg ml$^{-1}$ tricaine (Sigma-Aldrich), mounted in 2% methylcellulose and photographed with a Nikon fluorescent microscope (AZ100) equipped with a Nikon CCD camera (Qimaging Retiga 2000R). All the experiments involving zebrafish had been conducted according to the guidelines established by the Institutional Animal Care and Use Committee (IACUC) at The Scripps Research Institute, IACUC approval number 09-0009. Statistical analyses were performed with the software SPSS Statistics 19. The effects of different morpholinos on ISV development were analyzed with $\chi^2$ test.

## Acknowledgements

We would like to thank Professor Paul Schimmel for his advice on the project. This work is supported by the NIH grants GM088278 and NS085092 and a fellowship from the National Foundation for Cancer Research.

## Additional information

### Funding

| Funder | Grant reference number | Author |
|---|---|---|
| HHS | National Institutes of Health (NIH) | GM088278, NS085092 | Xiang-Lei Yang |

The funder had no role in study design, data collection and interpretation, or the decision to submit the work for publication.

### Author contributions

YS, Conception and design, Acquisition of data, Analysis and interpretation of data, Drafting or revising the article; XX, QZ, SK, Conception and design, Acquisition of data, Analysis and interpretation of data; GF, ZM, Acquisition of data, Analysis and interpretation of data; GSW, Acquisition of data; X-LY, Conception and design, Analysis and interpretation of data, Drafting or revising the article

### Ethics

Animal experimentation: All the experiments involving zebrafish had been conducted according to the guidelines established by the Institutional Animal Care and Use Committee (IACUC) at The Scripps Research Institute, IACUC approval number 09-0009.

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
