## [Decision Letter]

Thank you for sending your work entitled “Seryl-tRNA synthetase is essential in vascular development by recruiting SIRT2 to antagonize c-Myc” for consideration at *eLife*. Your article has been favorably evaluated by a Senior editor and 2 reviewers, one of whom is a member of our Board of Reviewing Editors.

The Reviewing editor and the other reviewer discussed their comments before we reached this decision, and the Reviewing editor has assembled the following comments to help you prepare a revised submission.

This manuscript reports on studies of mechanisms by which seryl-tRNA synthetase (SerRS) functions in the nucleus to regulate VEGFA expression. The major findings are 1) SerRS binds directly to the VEGF promoter at a site that overlaps with a binding site for cMyc. 2) In vivo and in vitro experiments provide evidence that SerRS can compete for cMyc for binding at these sites. 3) The authors confirm a prior proteomic result suggesting interaction of SerRS with SIRT2. Evidence is presented that is consistent with SerRS acting to recruit SIRT2 to the VEGFA promoter where it actively represses VEGFA expression via histone deacetylation. Loss of function experiments in zebrafish that focus on vascular branching are supportive of mutually antagonistic roles of cMyc and SerRS in controlling vascular development, at least in part through regulation of the expression of VEGFA.

One reviewer stated, 'Overall the findings are novel and interesting and suggest mutually antagonistic functions of SerRS and cMyc at the VEGF gene, and possibly other Myc target genes. SerRS could thus represent an alternative mechanism for negative regulation of Myc in addition to that provided by Mad repressors that form Mad/Max heterodimers'. The second reviewer stated, 'The findings are very novel, exciting, and important. The approach takes advantage of cell culture, in vitro, and zebrafish technologies, and is elegant, well-designed, and with appropriate controls.'

The reviewers also raised the following major concerns that would need to be adequately addressed for the manuscript to be accepted:

1) The evidence that SerRS binds to the VEGF promoter and antagonizes cMyc appears to be convincing. A major concern relates to the features of SerRS that enable it to bind to DNA in a sequence-specific manner. The authors make no mention of a conserved DNA binding domain in SerRS. Additional mapping studies are needed to define the amino acid sequences of SerRS required for sequence-specific DNA binding activity.

2) At present there does appear to be sequence specificity, but perhaps too much so, in that all point mutations shown in Figure 3 abolished binding and the minimal binding site was rather long. Are there base pairs in the binding site that are not important for sequence specific recognition? Are there mutations that discriminate binding of cMyc and SerRS?

3) The shifted bands in the EMSA in Figure 4 should be more precisely identified. For example, the band labeled as SerRS could be a “supershifted” complex containing SerRS as well as c-Myc-Max. Western blot analysis or antibody supershifts could clarify this issue. The authors should address why the Max/Max/DNA complex is not disrupted by SerRS.

4) In Figure 5 the authors show the interaction of SerRS with SIRT2 in an overexpression system only. The interaction between the endogenous proteins should be shown in cells, for example, by co-IP.

---

## [Author Response]

*1) The evidence that SerRS binds to the VEGF promoter and antagonizes cMyc appears to be convincing. A major concern relates to the features of SerRS that enable it to bind to DNA in a sequence-specific manner. The authors make no mention of a conserved DNA binding domain in SerRS. Additional mapping studies are needed to define the amino acid sequences of SerRS required for sequence-specific DNA binding activity*.

The reviewers made an important point. To address this point, we performed additional experiments to identify DNA binding sites on SerRS through domain mapping and deletion mutagenesis. The results indicate that SerRS has an extensive DNA-binding site. All three domains of SerRS (N-terminal tRNA binding domain, catalytic domain and the C-terminal UNE-S domain) are critical, but insufficient on their own, for DNA binding. To further define the DNA binding site on SerRS, we made 4 additional deletion mutants of SerRS at strategic positions. All 4 mutants lost the DNA binding activity. These results and the results on our investigation to address the second major concern of the reviewers (DNA sequence specificity; see below) now constitute a new section of the manuscript entitled “Characterize the interaction between SerRS and DNA” and the new Figure 4. These results also allow us to model the SerRS-DNA interaction, which is illustrated by a new movie (Video 1) added to the manuscript.

*2) At present there does appear to be sequence specificity, but perhaps too much so, in that all point mutations shown in*
Figure 3
*abolished binding and the minimal binding site was rather long. Are there base pairs in the binding site that are not important for sequence specific recognition? Are there mutations that discriminate binding of cMyc and SerRS?*

The answer is ‘yes’ to both questions. We now tested a total of 11 single or double mutations in the 27 bp DNA of the SerRS binding site. Two double mutants of the E-box, including one that would completely abolish c-Myc/Max binding, did not affect the SerRS interaction; on the other hand, 5 different single mutations outside the E-box (on both the 5’ and the 3’ sides) that would not affect c-Myc/Max binding greatly weakened SerRS binding, indicating that SerRS and c-Myc/Max have distinct DNA binding specificities. These new data is now added to the manuscript and is part of the new Figure 4.

*3) The shifted bands in the EMSA in*
Figure 4
*should be more precisely identified. For example, the band labeled as SerRS could be a “supershifted” complex containing SerRS as well as c-Myc-Max. Western blot analysis or antibody supershifts could clarify this issue. The authors should address why the Max/Max/DNA complex is not disrupted by SerRS*.

In parallel with the EMSA, we now performed the Western blot analysis as suggested to clarify this concern. Only SerRS was detected in the shifted band, indicating that SerRS and c-Myc/MAX cannot complex with the DNA simultaneously. This result is consistent with the result from the ChIP assay where SerRS and c-Myc were found to inhibit each other for binding to the VEGFA promoter. The result is now added to the manuscript as Figure 5—figure supplement 1.

The reason for why SerRS cannot disrupt MAX/MAX-DNA complex is because that the affinity between MAX homodimer and DNA is significantly higher (Kd = 19.2∼48.7 nM) than that of the SerRS-DNA interaction (Kd = 211.5∼265 nm) and the c-Myc/MAX-DNA interaction (Kd = 90.5∼229 nM). This information is now added to the manuscript.

*4) In*
Figure 5
*the authors show the interaction of SerRS with SIRT2 in an overexpression system only. The interaction between the endogenous proteins should be shown in cells, for example, by co-IP*.

We have performed the co-IP experiment to detect the interaction between the endogenous SerRS and SIRT2 as suggested. The experiment was successful and the new data are presented as Figure 6—figure supplement 1.